# Enhancement of Visible-Light Photocatalytic Degradation of Tetracycline by Co-Doped TiO_2_ Templated by Waste Tobacco Stem Silk

**DOI:** 10.3390/molecules28010386

**Published:** 2023-01-02

**Authors:** Quanhui Li, Liang Jiang, Yuan Li, Xiangrong Wang, Lixia Zhao, Pizhen Huang, Daomei Chen, Jiaqiang Wang

**Affiliations:** 1School of Materials and Energy, Yunnan Province Engineering Research Center of Photocatalytic Treatment of Industrial Wastewater, School of Chemical Sciences & Technology, School of Engineering, National Center for International Research on Photoelectric and Energy Materials, Yunnan University, Kunming 650091, China; 2Kunming Academy of Eco-Environmental Sciences, Kunming 650032, China

**Keywords:** visible-light photocatalysis, Co-doped TiO_2_, one-pot impregnation, tetracycline hydrochloride degradation, biotemplate

## Abstract

In this study, Co-doped TiO_2_ was synthesized using waste tobacco stem silk (TSS) as a template via a one-pot impregnation method. These samples were characterized using various physicochemical techniques such as N_2_ adsorption/desorption analysis, diffuse reflectance UV–visible spectroscopy, X-ray diffraction, field-emission scanning electron microscopy, high-resolution transmission electron microscopy, X-ray photoelectron spectroscopy, photoluminescence spectroscopy, and electron paramagnetic resonance spectroscopy. The synthesized material was used for the photodegradation of tetracycline hydrochloride (TCH) under visible light (420–800 nm). No strong photodegradation activity was observed for mesoporous TiO_2_ synthesized using waste TSS as a template, mesoporous Co-doped TiO_2_, or TiO_2_. In contrast, Co-doped mesoporous TiO_2_ synthesized using waste TSS as a template exhibited significant photocatalytic degradation, with 86% removal of TCH. Moreover, owing to the unique chemical structure of Ti-O-Co, the energy gap of TiO_2_ decreased. The edge of the absorption band was redshifted, such that the photoexcitation energy for generating electron–hole pairs decreased. The electron–hole separation efficiency improved, rendering the microstructured biotemplated TiO_2_ a much more efficient catalyst for the visible-light degradation of TCH.

## 1. Introduction

In recent years, tetracycline (TC) has been misused, and its residues have significantly impacted the ecosystem and the physical and mental health of human beings owing to its improper degradation treatment [1,2,3,4]. Therefore, an effective and environmentally friendly method is required for TC degradation. Photocatalysis is one of the most effective and economical methods for the removal of organic pollutants. Recently, novel photocatalysts Ag_3_PO_4_@MWCNTs@PPy and Ag_3_PO_4_@NC with excellent photocatalytic activity and photostability were successfully synthesized [5,6]. TiO_2_ is a widely used photocatalyst in this regard, owing to its low cost, low toxicity, and good stability [7,8]. However, TiO_2_ has a wide forbidden bandwidth and reacts only to UV light from sunlight. Moreover, the photogenerated electrons and holes easily recombine, limiting its widespread application. Therefore, it is necessary to develop an efficient photocatalyst that can utilize most of the light from the solar spectrum to effectively degrade TC. Ion doping or the use of biotemplates is a common approach to enhance the photocatalytic performance of TiO_2_. However, the effect of the simultaneous use of biotemplates and transition metal ion doping on the photocatalytic degradation efficiency has not been explored sufficiently.

Transition metal doping is an effective strategy for overcoming the limitations of TiO_2_, as it can improve light absorption and conductivity, reduce carrier complexation on TiO_2_ surface [9], and improve the photocatalytic performance of quantum-sized TiO_2_ by changing the band gap position and Fermi energy level [10]. The introduction of transition metals to improve the properties of TiO_2_ has been often reported; among them, Mn^2+^ [11], Co^2+^ [10], Fe^2+^ [12], and Ni^2+^ [13] plasma doping have been reported in detail. Among these ions, Co^2+^ is preferred because the ionic radius of Co^2+^ (0.74 Å) is similar to that of Ti^4+^ (0.61 Å) and the former can easily replace Ti^4+^ in TiO_2_ to form a stable structure. Doping with transition metal ions to change the crystal structure of TiO_2_ also alters the forbidden band width and cell parameters and improves the photocatalytic activity [14,15]. A series of Co-doped TiO_2_ materials synthesized by the sol–gel method were found to be effective for the visible-light degradation of methyl orange—increasing Co doping concentration enhanced the redshift of the UV–vis absorption spectrum. The dopant inhibited the growth of TiO_2_ grains, causing them to aggregate. This shifted the absorption maximum of TiO_2_ from the UV to visible region [16]. However, as electron complex centers, Co ions will decrease in the lifetime of the photogenerated electron–hole pairs. In this regard, controlling the valence state of Co ions to limit the utilization of electron–hole pairs is a more reasonable solution [17]. The present study was conducted to improve the photocatalytic efficiency by adjusting the Co^2+^/Co^3+^ ratio—a method that has rarely been reported.

The strategy of using natural biomass to modulate the morphology of prepared materials and produce nanoscale catalysts with specific functions has attracted considerable attention [18,19]. Many TiO_2_ materials with specific morphologies have been synthesized from biotemplates such as leaves, flower petals [20,21], foliage [22,23], bamboo [24], wood [25,26], and cotton [27]. Biotemplates can effectively improve the photocatalytic activity of TiO_2_. Moreover, materials synthesized using biotemplates exhibit better photocatalytic degradation efficiency toward pollutants. Tobacco stem silk (TSS) is coarse and tobacco leaves have stratified veins. Consequently, these plant parts, whose microstructures are in the form of lamellar folds, are mostly discarded nowadays.

In our previous study, we prepared an efficient TiO_2_ photocatalyst, TTS-ST(HF) [28], in which the morphology was modulated with TSS as a biotemplate. In this study, using tetrabutyl titanate (TBOT) and Co(NO_3_)_2_·6H_2_O as the Ti and Co sources, respectively, an efficient Co-doped TiO_2_ photocatalyst was synthesized using the one-pot impregnation method. TCH was used as a simulated pollutant to determine the photocatalytic degradation performance. The mechanism of heteroenergetic (Ti-O-Co) photocatalytic degradation has also been proposed.

## 2. Results

### 2.1. Synthesis of Photocatalysts by One-Pot Impregnation and Their Structural Characterization

In this study, we adopted a simple one-pot impregnation method (Figure 1), which is more concise than the conventional sol–gel biotemplate method, to replicate the complex structure of hard biotemplates in the microstructure of TiO_2_. Doped metal ions act as charge complex centers and decrease the lifetime of the electron–hole pairs. Hence, we designed a scheme to reduce the extent of conversion between Co^2+^ and Co^3+^ ions by controlling the amount of doped Co ions to determine the ratio of Co^2+^ ions to Co^3+^, thus enhancing the photocatalytic efficiency [17].

The N_2_ adsorption–desorption curves (Appendix A) of all prepared samples were characteristic type IV isotherms with a H3-type hysteresis loops, indicating the presence of mesopores with a stacked pore structure.

The X-ray diffraction (XRD) patterns (Figure 2) of the prepared materials were acquired to investigate the crystal structure of the biotemplated TiO_2_. The peaks at 2θ values of 25.34°, 37.01°, 37.85°, 38.64°, 48.14°, 53.97°, 55.18°, 62.24°, and 62.81° correspond to the (101), (103), (004), (112), (200), (105), (211), (213), and (204) crystal planes, respectively, consistent with standard card JCPDS 73-1764. The position of the characteristic diffraction peak of TiO_2_ did not change, and no dichotomous peak related to Co was observed, which could be attributed to the low Co content [29]. It is worth noting that TiO_2_ and the TiO_2_-TSS materials were purely white in color, and the color gradually changed to green with increasing Co doping. Moreover, the crystalline shape of TiO_2_ did not change after the introduction of the TSS filament template.

We next studied the morphology of the prepared samples. Figure 3 shows the SEM images of the pure TSS templates. It is evident that the stalk filaments have lamellar structures with folds, and their sizes range from 200 to 300 μm. Figure 4 shows the SEM and transmission electron microscopy (TEM) images of Co-TiO_2_/TSS(0.5) at different magnifications. Figure 4a–d indicates that Co is fully bound to the active sites on the surface of the TSS during the impregnation process and was successfully replicated after calcination at 450 °C, showing a lamellar folded morphology wherein the shrinkage and collapse of the pore structure resulted in size reduction after calcination at high temperatures [30]. Appendix A shows the SEM images of Co-TiO_2_/TSS(5.0). Appendix A shows that with increasing Co doping, the structure becomes denser and the active sites may not be sufficiently bound, leading to decreased photocatalytic efficiencies. The TEM images also show the microstructure block of the material. Figure 4e shows the lattice stripes and lattice spacing of the Co-TiO_2_/TSS(0.5). Analysis suggests that all the lattice spacings (d) are 0.351 nm, corresponding to the (101) crystal plane of the anatase crystal. Appendix A shows the TEM image of Co-TiO_2_/TSS(5.0); only TiO_2_ lattice stripes are seen despite increased Co doping, indicating that the material has only an anatase crystalline phase. This is consistent with the XRD analysis [31]. The microstructural morphology (Figure 4f) of Co-TiO_2_/TSS(0.5) shows closely spaced nanoparticles as microstructural building blocks that can exist in a lamellar state.

### 2.2. Analysis of Photocatalytic Activity

The photodegradation of TCH (13 mg/L) was used to assess the visible-light photocatalytic activity of the prepared samples. For comparison, pure TiO_2_ and Co-doped TiO_2_, prepared using a method similar to that for Co-TiO_2_/TSS, but without TSS, were used as the reference. As shown in Figure 5a, the removal rates with pure TiO_2_, TiO_2_-TSS, Co-TiO_2_/TSS(0.5), Co-TiO_2_/TSS(1.0), Co-TiO_2_/TSS(2.0), Co-TiO_2_/TSS(5.0), and Co-TiO_2_, as calculated according to equations (1) and (2), are 12%, 65%, 86%, 76%, 54%, 62%, and 44%, respectively. The photocatalytic degradation rates of TCH over pure TiO_2_, TiO_2_-TSS, Co-TiO_2_/TSS(0.5), Co-TiO_2_/TSS(1.0), Co-TiO_2_/TSS(2.0), Co-TiO_2_/TSS(5.0), and Co-TiO_2_ are 10%, 62%, 84%, 74%, 50%, 57%, and 52%, respectively. During the photocatalysis, no significant degradation of TCH was observed in the absence of catalysts under visible-light irradiation. The performance of the biotemplate-modified TiO_2_ was significantly improved, and the best results were obtained for the materials after modification with both Co ions and biotemplate. The best photocatalyst was found to be Co-TiO_2_/TSS(0.5), with 86% TCH removal after 90 min of visible-light irradiation.

Table 1 summarizes the comparison of the photodegradation efficiency of TCH overdifferent photocatalysts. Obviously, UV light or simulated solar (500 W Xeon lamb) were used in some investigations. Our biotemplated TiO_2_ was one of the efficient visible light photocatalysts.

The photocatalytic degradation of TCH follows first-order kinetics, which can be derived from Equation (1). Figure 5b,c suggests that the first-order kinetic constants of pure TiO_2_, TiO_2_-TSS, Co-TiO_2_, and Co-TiO_2_/TSS(X) are 0.001, 0.0102, 0.0196, 0.01442, 0.0074 0.0036, and 0.0088, respectively. Doping with an appropriate amount of Co ions can improve the photocatalytic efficiency of materials. Consistent with this, the photocatalytic efficiency of Co-TiO_2_/TSS(0.5) was 19 times higher than that of pure TiO_2_.

Usually, the efficient degradation of TCH by biotemplated TiO_2_ is achieved using UV irradiation [34]. Notably, the biotemplated photocatalyst prepared using the one-pot impregnation method can efficiently utilize the maximum percentage of the solar spectrum to degrade TCH.

### 2.3. Cyclic Stability Test

The stability experiments of Co-TiO_2_/TSS(0.5) were carried out (Appendix A). As shown in Appendix A, the removal rate decreased from 86% to 66% after 5 cycling runs. It can be due to the reduced adsorption rate in the dark adsorption process (Appendix A). After the fifth cycle, the adsorption rate is reduced from 12.2% to less than 5%. The reason could be explained that the adsorbed intermediate products may block the pores and occupy adsorption sites of catalyst. However, the total amount of TCH removed by Co-TiO_2_/TSS(0.5) for five cycling experiments was similar, which were 18.2, 19.2, 18.1, 16.9, and 16.7 mg/g, respectively. This result indicated that Co-TiO_2_/TSS(0.5) was a stable photocatalyst for TCH degradation. After 5 cycles of experiments, the efficiency dropped by 8.2%.

### 2.4. Identification of the Active Species and Elucidation of Mechanism

We designed an experiment to capture the active species in order to determine the main active species for the photocatalytic degradation of TCH. EDTA-2Na, BQ, and AgNO_3_ were used as trapping agents for h^+^, ·O_2_^−^, and e^−^, respectively. The results for the photocatalytic degradation of TCH by Co-TiO_2_/TSS(0.5) are shown in Figure 6a. When no trapping agent was added, the removal rate was 86%. When EDTA-2Na and BQ were added, the photocatalytic degradation rate decreased to 44.9% and 59.6%, respectively, indicating that the main active species were h^+^ and ·O_2_^−^. When AgNO_3_ was added, the catalytic efficiency increased to 100%. It is likely that the photogenerated electrons were trapped, promoting the effective separation of photogenerated electrons and holes and generating more h^+^, further indicating that h^+^ was the main active species [28].

To further understand the active species in the photocatalytic degradation of TCH by Co-TiO_2_/TSS(0.5), the presence of ·OH, h^+^, and·O_2_^−^ was detected by electron paramagnetic resonance (EPR) spectroscopy. TEMPO was used to capture h^+^ (Figure 6b), and a clear TEMPO signal was detected under dark conditions. When visible light was irradiated for 5 min, the signal from TEMPO significantly weakened, indicating the probable depletion of TEMPO due to h^+^. In addition, this indicated the generation of cavities under light irradiation. The presence of ·OH, h^+^, and O_2_^−^ was verified using DMPO (Figure 6c,d). No signal was detected under dark conditions, and weak signals from DMPO-OH and DMPO-O_2_^−^ were detected after 5 min of light irradiation. This indicated that the -OH and -O_2_^−^ active species were produced under light irradiation. Thus, the EPR experiments confirmed that these species played a role in the photocatalytic degradation. The dominant role in the photocatalytic degradation was played by h^+^ and ·O_2_^−^, which is consistent with the results of the active species-capture experiments.

### 2.5. Photocatalytic Reaction Mechanism

#### 2.5.1. X-ray Photoelectron Spectroscopy (XPS) Analysis

To analyze the chemical states of the sample surface, XPS analysis of the Co-TiO_2_/TSS(X) materials was performed. As shown in Figure 7a, Co-TiO_2_/TSS(X) consisted of four elements: C, Ti, O, and Co. The characteristic signal of Co was not obvious, probably because of its low content. However, the signal became stronger with increasing doping, and when Co: Ti ≥ 1, the peak intensity increased. Three chemical states of C can be observed in the high-resolution C 1 s spectra (Figure 7b). The peaks at 284.8 and 286.4 eV corresponded to carbon species present in the main chain and C-O bonds, attributable to C in indeterminate contaminants. Considering the presence of residual organic matter in the biotemplate [35], the species with a binding energy of 288.5 eV may be attributed to O-C=O, because the calcination of the remaining C species is incomplete [36]. Figure 7c shows a high-resolution O 1 s spectrum, with characteristic peaks of the Ti-O-Ti bond; that is, peak corresponding to lattice oxygen at 529.9 eV and that corresponding to the -OH group on the TiO_2_ surface at 531.9 eV out [37]. The binding energy peaks of O 1 s of Co-TiO_2_/TSS(0.5) appear at 529.32 and 530.86 eV, and the binding energy shifts slightly with increasing Co concentration, which may be due to the formation of the Ti-O-Co bonds [37,38]. Figure 7d shows the high-resolution Co 2p spectra. The Co 2p spectrum of CO-TiO_2_/TSS(0.5) shows two main peaks at 781.8 and 796.83 eV, corresponding to Co 2p_3/2_ and Co 2p_1/2_, respectively. The small difference between the binding energies (Δ = 15.7 eV) of the Co 2p_1/2_ and Co 2p_3/2_ orbitals indicates that high-spin Co^2+^ is essentially in the oxidation state, and the two main peak difference (Δ = 15 eV) indicates that the low-spin Co^3+^ is essentially in the oxidation state [39]. When the Co:Ti ratio was ≥1, two different Co peaks were observed. With increasing Co doping, the peak area of Co^3+^ increased, and the catalytic activity decreased. This could be attributed to the hybridization of the appropriate energy levels of (Ti-O-Co^3+^) and (Ti-O-Co^2+^) [40]. Co^3+^ can capture the electrons excited under light irradiation and reduce to Co^2+^ [41]. The adsorbed oxygen molecules on the TiO_2_ surface are reduced to ·O_2_^−^, following which Co^2+^ is oxidized to Co^3+^. However, excess Co in the material is detrimental to the photocatalytic efficiency because the metal ions act as charge complex centers and reduce the lifetime of the electron–hole pairs [17]. Figure 7e shows the high-resolution Ti 2p spectrum, where two characteristic peaks of Co-TiO_2_/TSS(0.5) Ti 2p_3/2_ and Ti 2p_1/2_, probably originating from spin-orbit splitting, can be observed [42]. The difference between the binding energies of Ti 2p_3/2_ and Ti 2p_1/2_ was 5.71 eV, consistent with previous reports [43,44]. The shoulder at 457.67 eV corresponds to Ti^3+^ of Ti_2_O_3_, and the slight shift in the binding energy and the shift in the intensity of the shoulder further indicate that the bandgap of Ti in the TiO_2_ matrix decreases with the substitution of Co [42]. Moreover, a decrease in the bandgap leads to a shift in the binding energy.

#### 2.5.2. Ultraviolet–Visible (UV–vis) Diffuse Reflectance Spectral Analysis

Figure 8 shows the UV–vis diffuse reflectance spectra of TiO_2_-TSS and Co-TiO_2_/TSS (X). The Co-TiO_2_/TSS(X) materials exhibited a higher light absorption ability than TiO_2_-TSS in the visible region, with redshifted absorption band edges. The enhanced visible-light absorption and narrower band gap energy can be attributed to the sensitization of biomass carbon dopants in the samples induced by the incomplete removal of the biotemplate [21]. The light absorption gradually became more robust with increasing Co doping. Figure 8b shows that the valence band position of Co-TiO_2_/TSS(0.5) is at 2.78 eV, indicating that Co doping has a negligible effect on the valence band position of TiO_2_, while the forbidden band width of Co-TiO_2_/TSS(0.5) was 3.01 eV. Figure 8a clearly shows that Co ions improve the photocatalytic activity by lowering the conduction band position, Ti-O-Co [45] chemical bond formation, which is consistent with the results of the XPS analysis.

#### 2.5.3. Photoluminescence (PL) Spectroscopy

To study the influence of the Ti-O-Co hybrid energy level formed upon photocatalysis and the characteristics of the photogenerated electron–hole pairs, we recorded the PL spectra and transient photocurrent response and performed electrochemical impedance spectroscopy (EIS) characterization of the materials. The separation efficiency of the photogenerated electrons, photogenerated electron–hole complexation, and migration efficiency of the modified TiO_2_-TSS and Co-TiO_2_/TSS(X) materials were determined from the PL spectra recorded at an excitation wavelength of 244 nm (Figure 9). As apparent from the figure, the higher the sample PL intensity, the higher the electron complexation efficiency [46]. The spectral intensity of the Co-TiO_2_/TSS(X) materials is much lower than that of TiO_2_-TSS, among which Co-TiO_2_/TSS(1.0) has the lowest spectral intensity. The photogenerated electron and holes were not easily combined, which is in good agreement with the experimentally obtained results of the photocatalytic activity. The presence of Co^3+^ is not conducive to photocatalysis; thus, the lesser the Co^3+^ content, the better will be the photocatalysis, because Co, as an electron complex center, will reduce the lifetime of the photogenerated electron–hole pair [47].

#### 2.5.4. Electrochemical Analysis

To further investigate the photogeneration of electrons and the transport of the synthesized photocatalysts under visible light, the transient photocurrent response and EIS spectra of TiO_2_, Co-TiO_2_/(5.0), TiO_2_-TSS, and Co-TiO_2_/TSS(0.5) were recorded. The transient photocurrent responses of pure TiO_2_, TiO_2_-TSS, and Co-TiO_2_/TSS(0.5) under visible-light irradiation are shown in Figure 10a. The transient photocurrent response of TiO_2_, TiO_2_-TSS, Co-TiO_2_/TSS(5.0), and Co-TiO_2_/TSS(0.5) increased sequentially, indicating an effective separation of the photogenerated electron and hole pairs of Co-TiO_2_/TSS(0.5).

The interfacial charge transfer of pure TiO_2_, TiO_2_-TSS, Co-TiO_2_/TSS(5.0), and Co-TiO_2_/TSS(0.5) was also investigated using chemical impedance spectroscopy (Figure 10b). The impedances of pure TiO_2_, TiO_2_-TSS, Co-TiO_2_/TSS(5.0), and Co-TiO_2_/TSS(0.5) decreased sequentially, indicating an increased charge transfer efficiency of Co-TiO_2_/TSS(0.5) in the photochemical system [48].

Our investigations reveal that the introduction of Co and regulation of the proportion of Co^2+^ to increase the lifetime of the photogenerated electron–hole pair can improve the photocatalytic activity.

#### 2.5.5. Elucidation of Photocatalytic Mechanism

Based on the analysis of our experimental results, a possible photocatalytic degradation mechanism was proposed. The Ti-O-Co hybridization energy level formed under visible-light irradiation reduced the forbidden bandwidth of TiO_2_, which was more favorable for electron excitation. During the catalytic process, a small amount of O_2_ dissolved in water reacts with the photogenerated electrons to produce a small amount of ·O_2_^−^ [49]. Moreover, a small number of photogenerated holes left in the valence band react with H_2_O to generate ·OH, which is responsible for the generation of -OH. The remaining holes, which are large in number, directly oxidize TCH, generating h^+^, ·OH, and ·O_2_^−^ as the final active species. A schematic of the photocatalytic mechanism is shown in Figure 11.

## 3. Materials and Methods

### 3.1. Materials

TSS (Kunming Cigarette Factory, Kunming, China), cobalt nitrate hexahydrate (Co(NO_3_)_2_·6H_2_O, Shanghai Titan, Shanghai, China), TBOT (Adamas, Beijing, China), anhydrous ethanol (Xilong Chemical, Guangdong, China), hydrochloric acid (HCl, Chuandong Chemical, Chongqing, China), TCH (Adamas, Beijing, China), glutaraldehyde (Xilong Chemical, Beijing, China), *p*-benzoquinone (BQ, Adamas, Beijing, China), ethylenediamine disodium acetate (Acros Organics, Beijing, China), and silver nitrate (Ag(NO_3_) Adamas, Beijing, China) were used for the experiments. All chemically synthesized photocatalysts were of analytical grade and used without further purification.

### 3.2. Preparation of Photocatalyst

To prepare the TSS biotemplate, TSS was pretreated by soaking it in 5% glutaraldehyde for 12 h and 5% HCI for 12 h, followed by gradient dehydration with ethanol. The dehydrated gradient material was dried overnight in an oven at 90 °C and then left to stand. Then, 2 g of the treated TSS was weighed in a 100 mL beaker, and 50 mL of ethanol was added to it, followed by the addition of 5 mL of TBOT and an appropriate amount of Co(NO_3_)_2_·6H_2_O for 24 h. This process controlled the Co:Ti molar ratio to 0.1, 0.5, 1, 2, and 5. The solution was then poured and subjected to hydrolysis in petri dishes for 24 h. The hydrolyzed material was calcined in a muffle furnace at 450 °C for 10 h (2 °C/min), following which the temperature was reduced to room temperature to obtain the final material. The resulting materials were named Co-TiO_2_/TSS(X) (X = 0.1, 0.5, 1, 2, 5); the material without Co doping was denoted as TiO_2_-TSS, and the material without a template was denoted as Co-TiO_2_.

### 3.3. Characterization of the Prepared Photocatalysts

To obtain the powder XRD (Rigaku TTRAX III) patterns, samples were scanned using CuKα radiation in the 2θ range of 20–80° at a rate of 10°/min. Field-emission scanning electron microscopy (FE-SEM, Nova NanoSEM 450, FEI, Eindhoven, Netherlands) and TEM (JEM-2100, Japan Electron Optics Laboratory CO, LTD, Tokyo, Japan) were used to analyze the morphological structures of the materials. The Brunauer–Emmett–Teller (BET, Micromeritics, Norcross, GA, USA) surface area was measured using a Micromeritics Tristar II 3020 surface area and porosity analyzer. Degassing was performed for 6 h before the analysis. The surface chemical state of the material was analyzed by XPS (Thermo Fisher Scientific K-Alpha^+^) using single Al Kα radiation. High-resolution XPS scans were recorded at a PE of 30 eV (step size: 0.1 eV). The UV–vis diffuse reflectance spectra were recorded on a Shimadzu UV-2600 spectrophotometer.

A standard three-electrode system was used to measure the photocurrent response (CHI 660E) and EIS profiles (Metrohm PGSTAT 302 N). The prepared sample, Pt wire, and saturated Ag/AgCl electrodes were used as the working, counter, and reference electrodes, respectively. An aqueous Na_2_SO_4_ solution (0.5 mol/L) was used as the electrolyte.

Photoluminescence (PL, Hitachi High-Tech, Tokyo, Japan) spectroscopy was performed on a Hitachi F-7000 fluorescence spectrometer at room temperature (frequency, 9.85 GHz; power, 20 mW; modulation frequency, 100 kHz). The EPR (Bruker BioSpin, Billetica Massachusetts, Germany) spectra were recorded on a Bruker EMXnano spectrometer.

### 3.4. Photocatalytic Activity

In this study, we chose a LED lamp as the visible light source (wavelength range from 420 to 800 nm). Compared with a 300 W Xenon lamp, a 5 W LED lamp consumes lesser energy and saves more green energy. Remarkably, the Co-doped mesoporous TiO_2_ templated by waste tobacco stem silk exhibited high photocatalytic activity under 5 W LED lamp irradiation. Thus, a 5 W LED lamp was used as visible light to study the photocatalytic degradation of TCH, the simulated pollutant. The dark reaction time was 60 min, and the adsorption rate was calculated after the attainment of the adsorption-desorption equilibrium. The light reaction using the 5 W LED lamp (420–800 nm) was allowed to proceed for 90 min. Samples were collected every 15 min—1 mL was withdrawn for each dark inverse and light reaction and filtered through a 0.45 μm aqueous membrane filter. The filtrate was used for high-performance liquid chromatography (HPLC, Agilent Series 1260 C). Appendix A shows that the HPLC peak area and TC concentration are linearly related. The peak area of TC in this experiment is the same as that of TCH; TCH and TC are primarily the same compound, except that TC is free of water molecules [28]. The equation of the calibration curve is C = 15389.4X−3444.5 (r^2^ = 0.99988), where C is the concentration of TC (0–20 mg/L) and X is the peak area. The removal rate and kinetic constants were calculated using Equations (1) and (2) [28].
(1)$$\mathrm{Removal}\%=\frac{C_{0}-C}{C_{0}}\times 100\%$$
(2)$$Ln\left(\frac{C_{e}}{C}\right)=kt$$

Here, *C_0_* is the initial TCH concentration, C is the instantaneous TCH concentration, and *C_e_* is the equilibrium concentration of TCH.

## 4. Conclusions

In this study, photocatalysts with different Co:Ti molar ratios were synthesized by a one-pot impregnation method using waste TSS as a template. The photocatalytic degradation performance of the Co-doped mesoporous TiO_2_ synthesized using waste TSS as a template was higher (86% removal) than those of mesoporous TiO_2_ synthesized using waste TSS as the template (65%), mesoporous Co-doped TiO_2_ (44%), and TiO_2_ (12%) for TCH removal. When a Ti-O-Co structure was formed, Co replaced Ti in the TiO_2_ lattice, although the crystalline shape of TiO_2_ did not change upon doping with Co. Moreover, the new energy level formed by Co was located above the valence band, which lowered the energy gap of TiO_2_ and redshifted the edge of the absorption band. This resulted in lower photoexcitation energy for electron–hole pair generation, higher electron–hole separation efficiency, and significantly higher photocatalytic activity. In conclusion, an inexpensive and stable photocatalyst has been developed to improve the efficiency of TCH degradation, and the synthetic strategy can also be extended to other transition metal-doped photocatalytic materials.

## Figures and Tables

**Figure 1 molecules-28-00386-f001:**
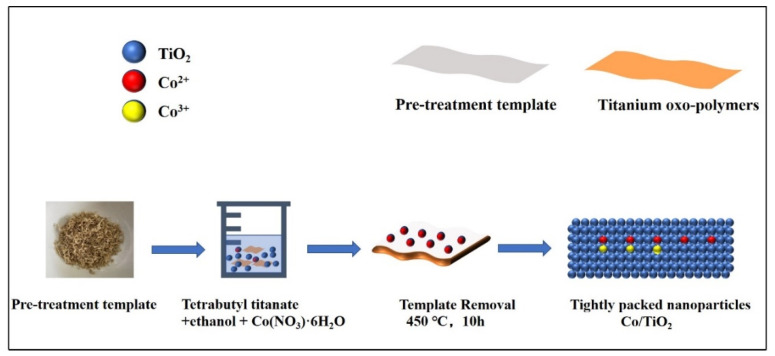
Schematic diagram of the synthesis through the one-pot impregnation method.

**Figure 2 molecules-28-00386-f002:**
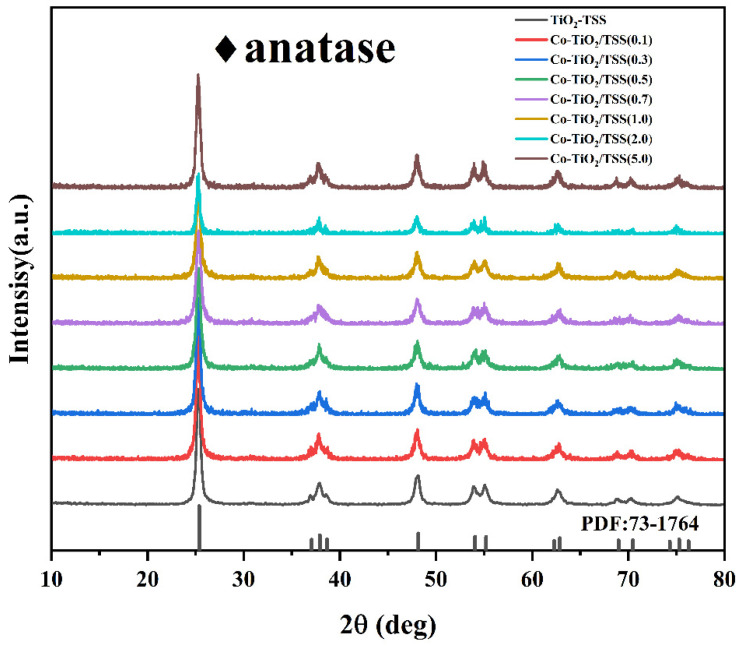
XRD patterns of Co-TiO_2_/TSS(X).

**Figure 3 molecules-28-00386-f003:**
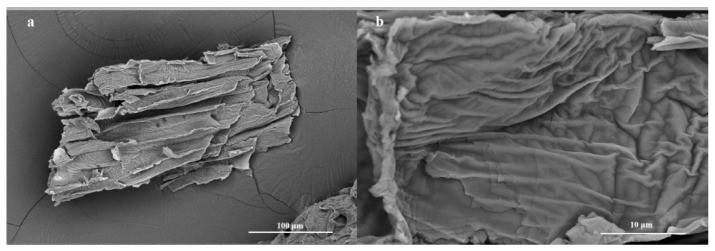
SEM images of original TSS at 1000× magnification (**a**) and 10,000× magnification (**b**).

**Figure 4 molecules-28-00386-f004:**
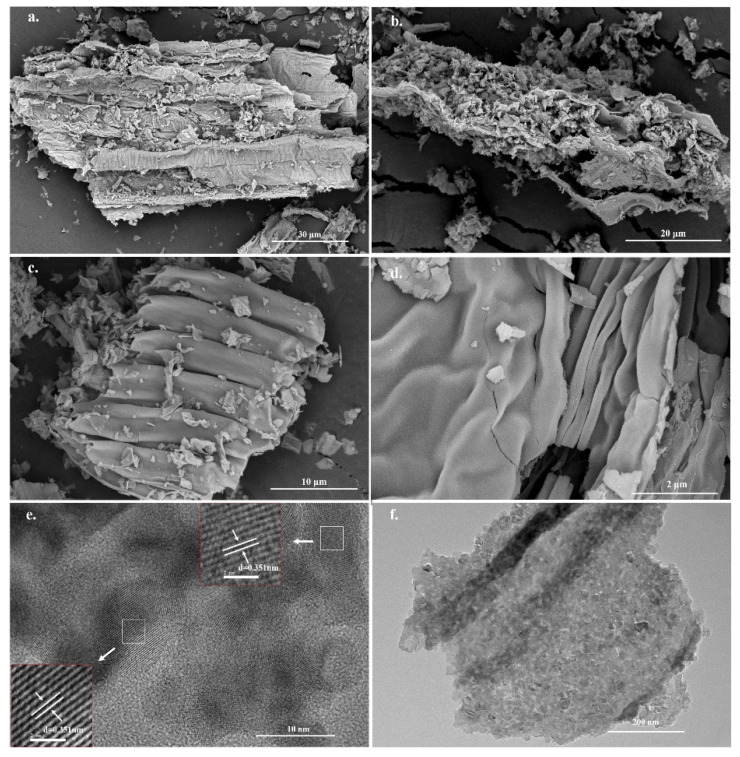
(**a**–**d**) SEM and (**e**,**f**) TEM images of Co-TiO_2_/TSS(0.5).

**Figure 5 molecules-28-00386-f005:**
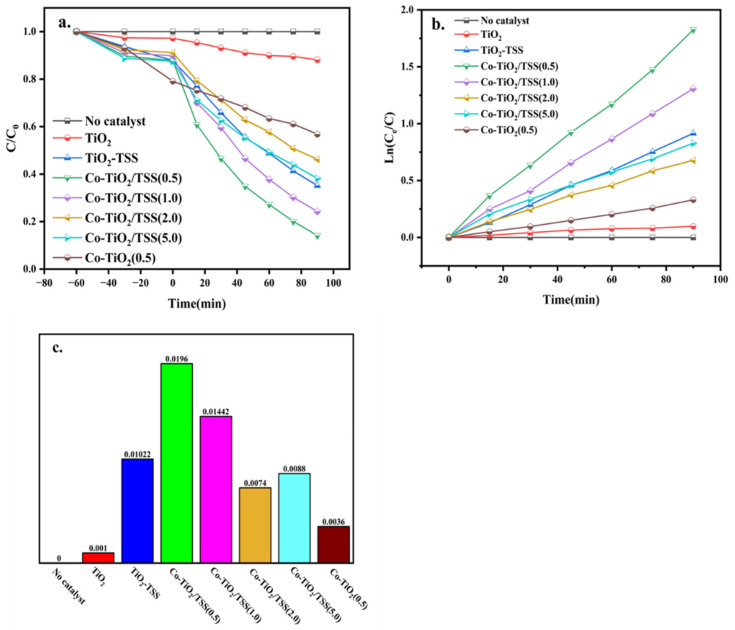
(**a**) TCH removal kinetics, (**b**) plot of ln (C_e_/C) versus irradiation time, (**c**) pseudo-first-order rate constants, k (min^−1^), for photocatalysis with pure TiO_2_, TiO_2_-TSS, Co-TiO_2_, and Co-TiO_2_/TSS(X).

**Figure 6 molecules-28-00386-f006:**
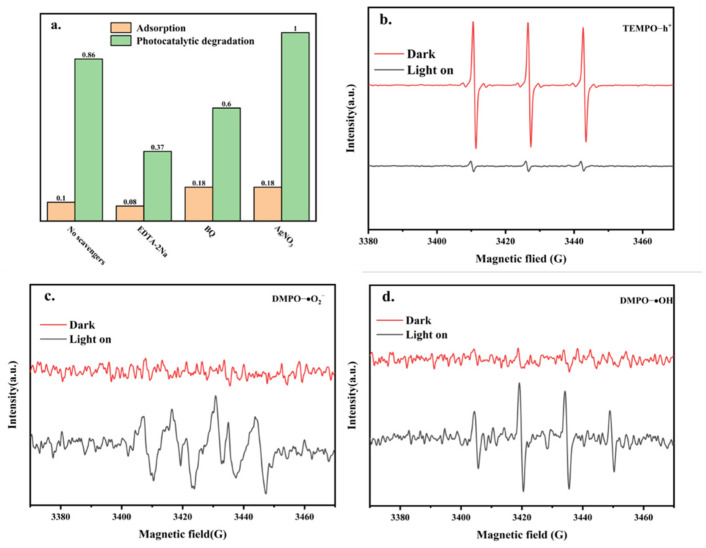
(**a**) Active species trapping experiments with Co-TiO_2_/TSS(0.5). EPR spectra of Co-TiO_2_/TSS(0.5) after the addition of (**b**) TEMPO to capture h^+^, (**c**) DMPO to capture OH, and (**d**) DMPO to capture O_2_^−^.

**Figure 7 molecules-28-00386-f007:**
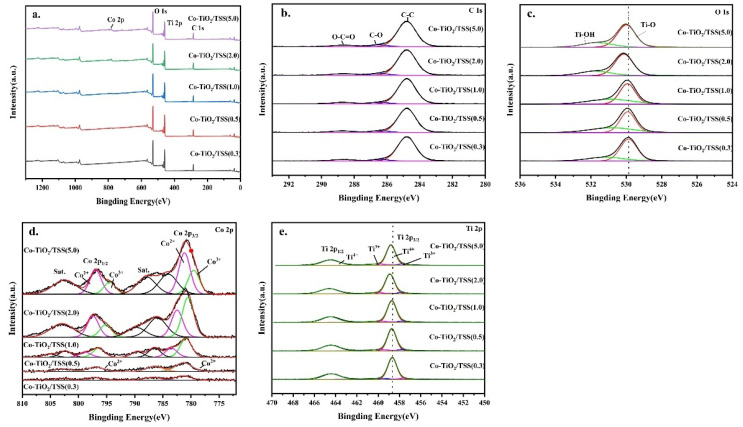
XPS spectra of (**a**) survey spectra, high-resolution XPS spectra of (**b**) C 1s, (**c**) O 1s, (**d**) Co 2p and (**e**) Ti 2p for as-prepared samples.

**Figure 8 molecules-28-00386-f008:**
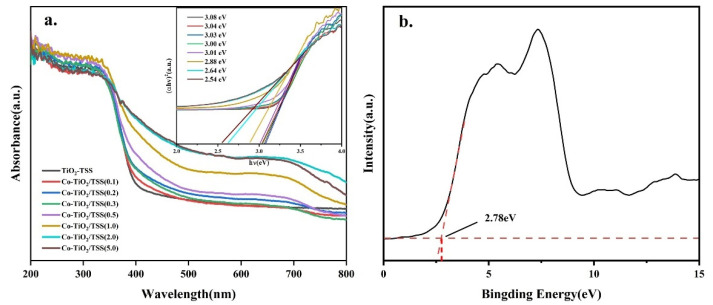
(**a**) UV–vis diffuse reflectance spectra of TiO_2_-TSS and Co-TiO_2_/TSS(X) and Tauc diagram (inset). (**b**) Valence band spectra of Co-TiO_2_/TSS(0.5).

**Figure 9 molecules-28-00386-f009:**
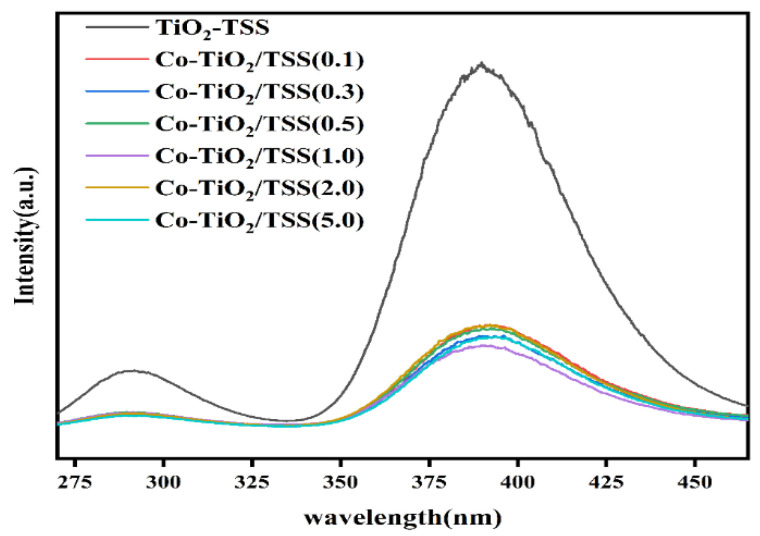
PL spectra of TiO_2_-TSS and Co-TiO_2_/TSS(X).

**Figure 10 molecules-28-00386-f010:**
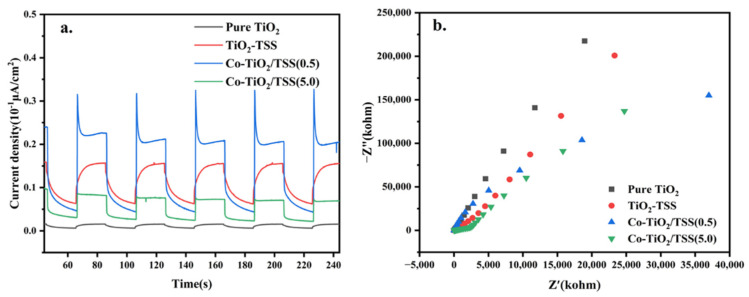
(**a**) Transient J-t photo response under visible-light illumination and (**b**) Nyquist plots of pure TiO_2_, TiO_2_-TSS, Co-TiO_2_/TSS(0.5), and Co-TiO_2_/TSS(5.0).

**Figure 11 molecules-28-00386-f011:**
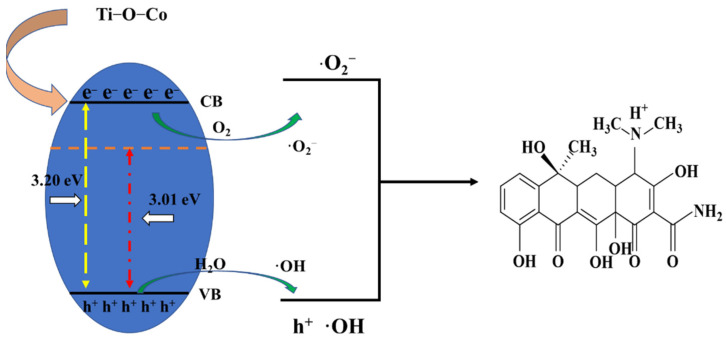
Reaction mechanism of visible-light-driven photocatalytic degradation of TCH on Co-TiO_2_/TSS(0.5).

**Table 1 molecules-28-00386-t001:** Comparison of photocatalytic activity for removal TCH over different photo catalysts.

Sample	Light Source and Irradiation Time	Initial Concentration and Catalyst Dosage	Degradation Rate (%)	Ref.
Ag_3_PO_4_@MWCNTs@PPy	300 W Xeon lamp, 2 min	20 mg/L 0.5 g/L	94.48	[5]
Cu_2_O-TiO_2_-Pal	500 W Xeon lamp, 4 h	30 mg/L 1.0 g/L	81.85	[32]
TiO_2-X_/ultrathin g-C_3_N_4_/TiO_2-X_	300 W Xeon lamp, 60 min	10 mg/L 1.0 g/L	87.70	[33]
Co-TiO_2_/TSS	5 W LED lamp, 90 min	13 mg/L 0.5 h/L	86.00	This work

## Data Availability

The datasets supporting the conclusions of this article are included within the article and its Appendix A.

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
