# Peer review of "Enhancement of Visible-Light Photocatalytic Degradation of Tetracycline by Co-Doped TiO2 Templated by Waste Tobacco Stem Silk"

_molecules, 2023, doi:10.3390/molecules28010386_

Round 1
Reviewer 1 Report
In the manuscript entitled “Enhancement of visible-light photocatalytic degradation of tetracycline by Co-doped TiO2 templated by waste tobacco stem silk” by Li et al, the authors successfully synthesized Co-doped TiO2 by a one-pot impregnation method using waste tobacco stem filaments as a template and applied the catalyst to the photodegradation of tetracycline hydrochloride under visible light. In summary, the subject is interesting and suitable for this journal, and the manuscript has sufficient experimental data. However, there are still some concerns and issues that need to be properly addressed before the possible publication in Molecules. I have listed the additional comments and suggestions below:
1. On Introduction: As the title of the manuscript is “Enhancement of visible-light photocatalytic degradation of tetracycline by Co-doped TiO2 templated by waste tobacco stem silk”, and there are also many discussions about the degradation of tetracycline and visible-light drived photocatalyst in the part of result and discussion, the authors should consider to describe more on the progresses in these fields. The following literature could serve these purposes on some aspects: Applied Catalysis B: Environmental, 2019, 258: 117969; Advanced Functional Materials. 2020, 30(38): 2002918. Also, the relationships between those progresses and this study should be commented better.
2. On 3.4 Photocatalytic activity: Why did the authors choose 5 W LED as the light source to simulate sunlight? According to reports, 300 W xenon lamp is used in most studies to simulate sunlight.
3. Lines 96-98: X-ray diffraction (XRD) plots of the prepared material (Figure 2). As described by the authors, the positions of the characteristic diffraction peaks of TiO2 did not change, and no diffraction peaks associated with Co appeared. The authors should explain why there is no characteristic peak of Co ?
4. Lines 133-135: The images and discussion analysis of removal rate and optical efficiency of pure TiO2, TiO2-TSS, Co-TiO2/TSS(X) and Co-TiO2 are a bit ambiguous, Please provide the calculation process of optical efficiency and make a detailed analysis
5. Lines 168-169: TEMPO was used to capture h+ (Figure 6(a)), and a clear TEMPO signal was detected under dark conditions. This described the contents of Figure 6(b), not Figure 6(a). The authors should edit this manuscript more thoroughly and carefully.
6. Lines 259-261: “The transient photocurrent response of TiO2, TiO2-TSS, Co-TiO2/TSS(5.0), and Co-TiO2/TSS(0.5) increased sequentially”, which is inconsistent with Figure 10 (a).
7. On 2.2 Photocatalytic activity analysis: The authors found that the best photocatalyst Co-TiO2/TSS(0.5) had a TCH removal rate of 86% after 90 min of visible light irradiation. In order for the readers to better evaluate the catalyst performance, the authors should compare the catalytic performance of Co-TiO2/TSS(0.5) with the catalytic performance of more other reported catalysts. The following papers are available for reference: Applied Catalysis B: Environmental, 2019, 245: 71-86; Advanced Functional Materials, 2022, 32(5): 2108814. The authors should discuss more reasons by comparing including the data mentioned here and highlighting the highlights of this article.
8. The stability of photocatalyst is another important index to evaluate the performance of catalyst, while this manuscript did not involve continuous cycling experiments. The catalyst recycling test shall be carried out for at least five cycles, and investigate whether the doped element Co will leak during use.
Author Response
Point 1: On Introduction: As the title of the manuscript is “Enhancement of visible-light photocatalytic degradation of tetracycline by Co-doped TiO2 templated by waste tobacco stem silk”, and there are also many discussions about the degradation of tetracycline and visible-light drived photocatalyst in the part of result and discussion, the authors should consider to describe more on the progresses in these fields. The following literature could serve these purposes on some aspects: Applied Catalysis B: Environmental, 2019, 258: 117969; Advanced Functional Materials. 2020, 30(38): 2002918. Also, the relationships between those progresses and this study should be commented better.
Response 1: Thank you for your comments. We have added specific progress in the field of TCH degradation in the introduction and cited relevant references. Lines 36-38 in the text have been specifically shown, and We have marked them in red. The reference cited is No. 5 and No. 6.
Point 2: On 3.4 Photocatalytic activity: Why did the authors choose 5 W LED as the light source to simulate sunlight? According to reports, 300 W xenon lamp is used in most studies to simulate sunlight.
Response 2: Thank you for your suggestions. In this study, we choose a LED lamp as the visible light source (wavelength range from 420 to 800 nm). Compared with a 300 W Xenon lamp, a 5 W LED lamp consumes lesser energy and saves more green energy. Remarkably, the Co-doped mesoporous TiO2 templated by waste tobacco stem silk exhibited high photocatalytic activity under 5 W LED lamp irradiation. Thus, a 5 W LED lamp was used as visible light to study the photocatalytic degradation of TCH, the simulated pollutant. We have marked it in red in lines 353-358.
Point 3: Lines 96-98: X-ray diffraction (XRD) plots of the prepared material (Figure 2). As described by the authors, the positions of the characteristic diffraction peaks of TiO2 did not change, and no diffraction peaks associated with Co appeared. The authors should explain why there is no characteristic peak of Co?
Response 3: Thank you for your suggestions. The main reason why there is no characteristic peak related to Co is that the content of Co is low. We have marked it in red in lines 100-102 of the text.
Point 4: Lines 133-135: The images and discussion analysis of removal rate and optical efficiency of pure TiO2, TiO2-TSS, Co-TiO2/TSS(X), and Co-TiO2 are a bit ambiguous, Please provide the calculation process of optical efficiency and make a detailed analysis.
Response 4: Your suggestion is very effective. As shown in Figure 5(a), the removal rates with pure TiO2, TiO2-TSS, Co-TiO2/TSS(0.5), Co-TiO2/TSS(1.0), Co-TiO2/TSS(2.0), Co-TiO2/TSS(5.0), and Co-TiO2, as calculated according to formula (1-1,1-2), are 12%, 65%, 86%, 76%, 54%, 62%, and 44%, respectively. The photocatalytic degradation rates of TCH over pure TiO2, TiO2-TSS, Co-TiO2/TSS(0.5), Co-TiO2/TSS(1.0), Co-TiO2/TSS(2.0), Co-TiO2/TSS(5.0), and Co-TiO2 are 10%, 62%, 84%, 74%, 50%, 57%, and 52%, respectively. We have marked in red lines in 135-140 of the text.
Point 5: Lines 168-169: TEMPO was used to capture h+(Figure 6(a)), and a clear TEMPO signal was detected under dark conditions. This described the contents of Figure 6(b), not Figure 6(a). The authors should edit this manuscript more thoroughly and carefully.
Response 5: This error was caused by our carelessness and We have fixed it.
Point 6: Lines 259-261: “The transient photocurrent response of TiO2, TiO2-TSS, Co-TiO2/TSS(5.0), and Co-TiO2/TSS(0.5) increased sequentially”, which is inconsistent with Figure 10(a).
Response 6: Thank you for your suggestions. It should be that the transient photocurrent response of TiO2, Co-TiO2/TSS(5.0), TiO2-TSS, and Co-TiO2/TSS (0.5) increases sequentially. In line 282, we have marked it in red.
Point 7: On 2.2 Photocatalytic activity analysis: The authors found that the best photocatalyst Co-TiO2/TSS(0.5) had a TCH removal rate of 86% after 90 min of visible light irradiation. In order for the readers to better evaluate the catalyst performance, the authors should compare the catalytic performance of Co-TiO2/TSS(0.5) with the catalytic performance of more other reported catalysts. The following papers are available for reference: Applied Catalysis B: Environmental, 2019, 245: 71-86; Advanced Functional Materials, 2022, 32(5): 2108814. The authors should discuss more reasons by comparing including the data mentioned here and highlighting the highlights of this article.
Response 7: Thank you for your suggestions. Table 1 summarizes different photocatalysts for the degradation of TCH. The reported photocatalysts for efficient TCH degradation still needs to be implemented by using UV light or simulated solar (500 W Xeon lamb). Significantly, the novel biotemplated TiO2 provided by this work was the efficient visible light photocatalysts, which can both efficiently utilize the most percentage of the solar spectrum and degrade TCH.
Table 1. Comparison of photocatalytic activity for removal TCH over different photocatalysts
|
Sample |
Light source and Irradiation Time |
Initial concentration and catalyst dosage |
degradation rate (%) |
Ref |
|
Ag3PO4@MWCNTs@PPy |
300 W Xeon lamp, 2min |
20mg/L 0.5g/L
|
94.48 |
[1] |
|
Cu2O-TiO2-Pal |
500 W Xeon lamp, 4h |
30mg/L 1.0g/L
|
81.85 |
[2] |
|
TiO2-X/ultrathin g-C3N4/TiO2-X |
300 W Xeon lamp, 60 min |
10mg/L 1.0g/L
|
87.7 |
[3] |
|
Co-TiO2/TSS |
5 W LED lamp,90 min |
13mg/g 0.5h/L |
86 |
This work |
Point 8: he stability of photocatalyst is another important index to evaluate the performance of catalyst, while this manuscript did not involve continuous cycling experiments. The catalyst recycling test shall be carried out for at least five cycles, and investigate whether the doped element Co will leak during use.
Response 8: Thank you for your suggestions. The stability experiments of Co-TiO2/TSS(0.5) were carried out (Figure S5). As shown in Figure S5(a), the removal rate decreased from 86% to 66% after 5 cycling runs. It can be due to the reduced adsorption rate in the dark adsorption process (Fig. S5b). After the fifth cycle, the adsorption rate is reduced from 12.2% to less than 5%. The reason could be explained that the adsorbed intermediate products may block the pores and occupy adsorption sites of catalyst. However, the total amount of TCH removed by Co-TiO2/TSS(0.5) for five cycling experiments was similar, which were 18.2, 19.2, 18.1, 16.9, and 16.7 mg/g, respectively. This result indicated that Co-TiO2/TSS(0.5) was a stable photocatalyst for TCH degradation. After 5 cycles of experiments, the efficiency dropped by 8.2%.
Figure S5. (a) Removal rate of TCH, (b) adsorption rate of TCH, and (c) rate of total TCH removal over Co-TiO2/TSS under visible light during the cycling experiments.
References:
[1] Lin Y, Wu X, Han Y, et al. Spatial separation of photogenerated carriers and enhanced photocatalytic performance on Ag3PO4 catalysts via coupling with PPy and MWCNTs [J]. Applied Catalysis B: Environmental, 2019, 258(117969).
[2] Shi Y, Yang Z, Wang B, et al. Adsorption and photocatalytic degradation of tetracycline hydrochloride using a palygorskite-supported Cu2O–TiO2 composite [J]. Applied Clay Science, 2016, 119(311-20).
[3] Ni J, Wang W, Liu D, et al. Oxygen vacancy-mediated sandwich-structural TiO(2-x) /ultrathin g-C(3)N(4)/TiO(2-x) direct Z-scheme heterojunction visible-light-driven photocatalyst for efficient removal of high toxic tetracycline antibiotics [J]. J Hazard Mater, 2021, 408(124432).

Reviewer 2 Report
The authors showed the Co-doped TiO2 can be efficiently used for the photodegradation of tetracycline hydrochloride under visible light (420–800 nm). Overall, I suggest that it can be published after some minor modifications.
1. The effect of cobalt content should be investigated.
2. The Figure 4 and Figure 5 are not clear and should be redrawn.
Author Response
Point 1: The effect of cobalt content should be investigated.
Response 1: Thank you for your suggestions. We have explained the influence of Co content and marked lines 274-277 with red.
Point 2: Figure 4 and Figure 5 are not clear and should be redrawn.
Response 2: Thank you for your suggestions. We have modified Figure 4 and Figure 5.
Figure 4. (a-d) SEM and (e-f) TEM images of Co-TiO2/TSS(0.5).
Figure 5. (a) TCH removal kinetics, (b) plot of ln (Ce/C) versus irradiation time, (c) pseudo-first-order rate constants, k (min-1), for photocatalysis with pure TiO2, TiO2-TSS, Co-TiO2, and Co-TiO2/TSS(X).
